

# Molecular clocks, biogeography and species diversity in *Herichthys* with evaluation of the role of Punta del Morro as a vicariant brake along the Mexican Transition Zone in the context of local and global time frame of cichlid diversification

Fabian Pérez-Miranda[1],  Omar Mejia[1],  Benjamín López[1] and  Oldřich Říčan[2]

[1] Laboratorio de Variación Biológica y Evolución, Departamento de Zoología, Escuela Nacional de Ciencias Biológicas, Instituto Politécnico Nacional, Mexico City, Mexico

[2] Departament of Zoology, Faculty of Science, University of South Bohemia, České Budějovice, Czech Republic

Corresponding author
Omar Mejia, hmejiag@ipn.mx,
homarmejia@hotmail.com

## ABSTRACT

Using molecular dated phylogenies and biogeographic reconstructions, the species diversity, biogeography and time frame of evolution of the genus *Herichthys* were evaluated. In particular, we test the role of Punta del Morro (PdM) as a vicariant brake along the Mexican Transition Zone in the context of local and global time frame of cichlid diversification using several sets of calibrations. Species diversity in *Herichthys* is complex and the here employed dating methods suggest young age and rapid divergence for many species while species delimitation methods did not resolve these young species including both sympatric species pairs. Based on our molecular clock dating analyses, *Herichthys* has colonized its present distribution area significantly prior to the suggested vicariance by PdM (10–17.1 Ma vs. 5 to 7.5 Ma). The PdM constraint is in conflict with all other paleogeographic and fossil constraints including novel ones introduced in this study that are, however, congruent among each other. Our study demonstrates that any cichlid datings significantly older or younger than the bounds presented by our analyses and discussion have to be taken as highly questionable from the point of view of Middle American paleogeography and cichlid biogeography unless we allow the option that cichlid biogeography is completely independent from ecological and geological constraints.

## INTRODUCTION

*Myers (1966)*, in his seminal work, concluded that the fish fauna of Central America is dominated by secondary freshwater fishes (mainly Poeciliidae and Cichlidae) and that the fauna was established by colonization during the Early Tertiary, while primary fishes did

not arrive until a connection was established with South America in the Late Tertiary. Since Myers, several studies have tested and refined his hypothesis. *Rosen (1975)* put forth that the first connection for freshwater fishes between Middle America and South America might have started already in the late Cretaceous. However, for both the poeciliid and cichlid families, we still do not know their precise dates of colonization because the numerous studies dedicated to the topic have supported strongly different timeframes for their colonization. All studies agree that unlike the multiple widely spaced colonizations by the poeciliids, cichlids colonized Middle America during a single timeframe (*Hrbek, Seekinger & Meyer, 2007*; *Říčan et al. (2013)*; *Tagliacollo et al., 2017*; *Matamoros et al., 2015*). The first studies of cichlids used a 1% to 2% substitution rate of mitochondrial DNA, positing that colonization occurred between 11.3 to 13.3 Ma (*Martin & Bermingham, 1998*) and 14 to 18 Ma (based on a 1% substitution rate; *Concheiro-Pérez et al., 2007*). Using the Punta del Morro (PdM) as a calibration point (7.5 Ma), *Hulsey et al. (2004)* suggested a similar age of 16.2 Ma for colonization. However, using two geological events, the separation of Cuba and Hispaniola (23 to 24 Ma) and that of the Orinoco and Magdalena basins (10.1–11.8 Ma), *Concheiro-Pérez et al. (2007)* suggested an older age of 20 to 24 Ma for colonization. In the most detailed study of the cichlids so far, *Říčan et al. (2013)* employed as calibration points the same events as *Concheiro-Pérez et al. (2007)* plus the minimum age of the fossil, *Plesioheros chauliodus* (39.9–48.6 Ma), and concluded that the cichlid fishes colonized simultaneously Middle America and the Greater Antilles much earlier, between 32 and 24 Ma in the Oligocene. *Tagliacollo et al. (2017)* using a different set of fossil calibrations (plus a questionable minimum age of cichlids) reconstructed an even earlier colonization of Middle America and the Greater Antilles by the cichlids during the Paleocene-Eocene (ca. 45–50 Ma). Additionally, there are several other studies that include some Middle American cichlids (but are not dedicated to the Middle American and Caribbean cichlids) in dated phylogenies and these also provided a wide range of dates for the colonization. For example, the dated phylogeny of *Musilová et al. (2015)* suggested a colonization date of Middle America between 32 and 26 Ma, while the largest study dedicated to cichlid dating of *Matschiner et al. (2017)* suggested a colonization date between 44 and 31 Ma. Finally, the third group of studies used for calibration the assumed cichlid vicariance of western Gondwana (South America-Africa) and place the colonization around 55 Ma (e.g., *López-Fernández et al., 2013*).

Both the dedicated and non-dedicated calibrated phylogenies have thus provided a wide range of dates for the colonization of Middle America and the Antilles ranging from 16 Ma (*Hulsey et al. (2004)*, between 24 and 32 Ma *Concheiro-Pérez et al., 2007*; *Říčan et al., 2013*; *Musilová et al., 2015*), around 44 to 45 Ma (*Matschiner et al., 2017*; *Tagliacollo et al., 2017*) and up to 55 Ma (e.g., *López-Fernández et al., 2013*). The 24–32 Ma group of studies is compatible with the GAARlandia landbridge hypothesis of land connection between South America, the Greater Antilles and Middle America (*Iturralde-Vinent & MacPhee, 1999*) with only minor sea crossings, while the studies with the older dates imply or even advocate almost complete transoceanic dispersal, additionally complicated by the absence of terrestrial habitats in the Greater Antilles between the end of the Cretaceous and Eocene-Oligocene (ca. 66 and 37 Ma; *Iturralde-Vinent & MacPhee, 1999*;

*Macphee & Iturralde-Vinent, 2005*). Southern Central America, including Eastern Panama, is unequivocally reconstructed in all studies as having been colonized from Northern Middle America starting in the Early-Middle Miocene and a limited colonization of Eastern Panamá coincided with the first wave of colonization of this area by primary freshwater fishes from South America (*Bermingham & Martin, 1998*).

The available studies have thus given a very wide window of reconstructed dates of colonization of Middle America by the Cichlidae. Most of the debate in dating of cichlid phylogenies revolves around the topic of colonization of South America from Africa (review in *Matschiner, 2019*), a very ancient and controversial event. We believe that a better way by which to make progress in this debate is to concentrate on and better constrain the dating by focusing on the study of more regional, recent and hence tractable biogeography of cichlid fishes, in the case of this study within Middle America. The phylogeny of Middle American cichlids is now completely known based on complete species-level sampling and also genomic phylogenies (*Říčan et al., 2013*; *Říčan et al., 2016*; *Ilves, Torti & López-Fernández, 2018*). The testing of regional biogeographic patterns and events is thus straightforward using these datasets. Thus far, only one calibration point derived from Middle American biogeography was employed in the dating of cichlid biogeography, and that is the PdM (*Hulsey et al., 2004*; *Hulsey, Hollingsworth & Fordyce, 2010*). The PdM is a lava flow from the Transmexican Volcanic Belt that extends almost to the coastal line in the Mexican state of Veracruz, found at the northern border of the Neotropical zone where it terminates along a transition zone in Mexico.

The PdM has been demonstrated to act as an effective barrier or filter within this transition zone for freshwater species. The fish fauna drastically changes from being 75% of Nearctic origin north of PdM to 95% of Neotropical origin south of PdM (*Obregón-Barboza, Contreras-Balderas & Lozano-Vilano, 1994*; *Miller, Minckley & Norris, 2005*). The importance of the PdM as a biogeographic break has been recognized not only for aquatic species, but also reptiles and mammals (*Pérez-Higareda & Navarro, 1980*; *Mulcahy & Mendelson III, 2000*; *Savage & Wake, 2001*).

While the biogeographic break at the PdM is well-demonstrated, the historical influence of this postulated barrier is not understood. Most studies of freshwater fishes in this area assumed that the PdM acted as a vicariant event (in cichlids, characids and poecilids; *Hulsey et al., 2004*; *Mateos, 2005*; *Ornelas-García, Domínguez-Domínguez & Doadrio, 2008*; *Agorreta et al., 2013*; *Culumber & Tobler, 2016*; *Palacios et al., 2016*). However, the divergence time estimates in the aforementioned works (which range between 4.4 to 6 Ma) contrast strongly with the at least 14 Ma (and significantly more in several of the studies) estimated in other studies of cichlids, thereby clearly predating the formation of PdM (*Říčan et al., 2013*; *Říčan et al., 2016*; *Musilová et al., 2015*; *Tagliacollo et al., 2017*; *Matschiner et al., 2017*).

*Huidobro et al. (2006)* analyzed the distribution patterns of several groups of aquatic taxa distributed below 1,000 m altitude (crustaceans, angiosperms and fishes) and found three biogeographic tracks. One of them included the transition zone with the panbiogeographic track being interrupted at the PdM. Their study suggested that the track is coupled with eustatic sea changes that occurred during the Miocene, which is in agreement with the

older dating of fish divergences at the PdM (*Říčan et al., 2013*; *Říčan et al., 2016*; *Musilová et al., 2015*; *Tagliacollo et al., 2017*; *Matschiner et al., 2017*).

Among the cichlids only one genus, *Herichthys*, is found north of the PdM. Its phylogenetic position among the cichlids is now well-understood (*Říčan et al., 2013*; *Říčan et al., 2016*), but the reasons for it being the only genus north of the PdM are not agreed upon, similarly as the controversial cases of South American colonization and of colonization of Middle America. One group of studies maintains that this distribution is a result of vicariance caused by the formation of the PdM (*Hulsey et al., 2004*) as had been suggested for other fish groups (*Mateos, 2005*; *Ornelas-García, Domínguez-Domínguez & Doadrio, 2008*; *Agorreta et al., 2013*; *Culumber & Tobler, 2016*; *Palacios et al., 2016*). However, other studies put forth that colonization took place during the Miocene (*Říčan et al., 2013*) and hence significantly predated the origin of the PdM (*Říčan et al., 2013*; *Říčan et al., 2016*; *Musilová et al., 2015*; *Tagliacollo et al., 2017*; *Matschiner et al., 2017*). *Herichthys* is thus the only cichlid group able to shed light on cichlid divergence across the PdM. The clarification of this and other biogeographic events within Middle America in turn has the potential to narrow the timeframe of cichlid colonization of Middle America and of South America.

*Herichthys* is a unique genus among Middle American cichlids because it is the only cichlid genus in Middle America not found in sympatry with other genera and it is the northernmost cichlid genus on the Atlantic slope of the Americas (*Říčan et al., 2016*) as well as the only one present north of the Mexican Transition Zone. As a consequence of its isolation, *Herichthys* is the most diverse genus in terms of its ecomorphology among the Middle American cichlids (*Říčan et al., 2016*). In *Herichthys*, there are eco-morphologically polymorphic species, probably in early stages of speciation, and also sympatric and syntopic sister species that together with the nature of their isolation and ecomorphological divergence suggest sympatric speciation (*Říčan et al., 2016*; *Pérez-Miranda et al., 2018*). A large proportion of the diversity in *Herichthys* is therefore potentially the result of ecological opportunity afforded to the genus by the absence of other sympatric cichlid genera (*Seehausen, 2015*; *Říčan et al., 2016*; *Burress et al., 2018*; *Piálek et al., 2019*; *Piálek et al., 2018*).

*Herichthys* is thus of great promise as a model for the understanding of several general aspects of cichlid diversification in Middle America. Indeed, one of its species (*H. minckleyi*) has already become a model species owing to the presence of three distinct ecomorphs within this species. A similar degree of intraspecific diversity is also found in *H. bartoni*, where it however still needs to attract attention of the research community. *Herichthys* holds great potential to become a similarly useful model of diversification in North America, just as afforded by the genus *Amphilophus* in Central America, because of the many parallels between the two genera, most importantly diversification in both riverine and lacustrine habitats, sympatric ecological divergence, incipient speciation, and parallel evolution of novel traits (*Barluenga et al., 2006*; *Elmer et al., 2010a*; *Elmer et al., 2010b*; *Elmer et al., 2014*).

While the phylogenetic position of *Herichthys* within Middle American cichlids is well-established (*Říčan et al., 2016*), its species diversity is complicated and remains contested

(reviewed in *Pérez-Miranda et al., 2018*). *Pérez-Miranda et al. (2018)* have evaluated the usefulness of several nDNA and mtDNA markers and concluded that the best resolution of both phylogenetic relationships and species-diversity issues was achieved by the nDNA ddRAD and mtDNA cytb datasets among those currently available. Based on the available data that also included morphological diagnosability of the species, *Pérez-Miranda et al. (2018)* have concluded that *Herichthys* includes 11 species (one species has not been evaluated, *H. molango*, and several recently described species were placed in synonymy).

In this study, we first focus on the species diversity analyses within *Herichthys* using an extended sampling of the cytb marker (including *H. molango*) in order to clarify the diversity within *Herichthys*. We then use biogeographical analyses and dated phylogenies of *Herichthys* calibrated by various constraints to evaluate the PdM biogeographical break within the wider Middle American and global cichlid diversification context. Middle America is particularly useful for constraining cichlid datings due to its complex history and biogeography as an evolving bridge between two continents. Our approach should thus beside the clarification of *Herichthys* dating and biogeography serve to narrow the time window of cichlid colonization of Middle America, in particular, and of the Neotropics in general.

## MATERIAL AND METHODS

### Species identification, molecular dataset and laboratory protocols

All specimens analyzed in this study were euthanized according with the procedure described in the Mexican law NOM-033-SAG/ZOO-2014. We combine morphological species determination of *Herichthys* species with *post-hoc* species delimitation using the molecular mtDNA cytochrome b (*cytb*) marker. Specimens were identified to species with the use of original descriptions, identification keys, and comparative material. Our sampling for molecular analyses includes all described species of *Herichthys* including *H. molango* that has not been previously evaluated. The dataset is based on that of *Říčan et al. (2016)* and *Pérez-Miranda et al. (2018)* with the inclusion of many newly generated sequences (see supplemental Table S1). The dataset includes the complete cytb marker (1,137 bp) with 164 specimens (i.e., 141 haplotypes) from 62 localities (Fig. 1; compared to 99 specimens from 32 localities in *Říčan et al., 2016*; *Pérez-Miranda et al., 2018*). The dataset is anchored by 13 additional species, i.e., *Hypselecara coryphaenoides*, *Symphysodon aequifasciata*, *Mesonauta festivus*, *Heros* sp., *Uaru amphiacanthoides*, *Nandopsis tetracanthus*, *Nandopsis haitiensis*, *Caquetaia* sp. cf *kraussi*, *Caquetaia spectabilis*, *Vieja maculicauda*, *Vieja melanura*, *Thorichthys helleri* and *Thorichthys pasionis* which are used to provide an evenly spaced set of successive outgroups some of which are used for the various molecular clock dating analyses.

Genomic DNA was extracted from ethanol-preserved muscle tissue using a salt extraction protocol (*Aljanabi & Martinez, 1997*). The complete sequence of mitochondrial cytochrome b was amplified using the primers, CYTBF (5′AATGACTTGAAAAACCACCGTTG3′) and CYTBR (5′GTCTTGTAAACCGGACGCCGAA 3′), designed for this study. PCR amplifications were performed in a thermocycler (Geneamp PCR System 9700, Carlsbad,

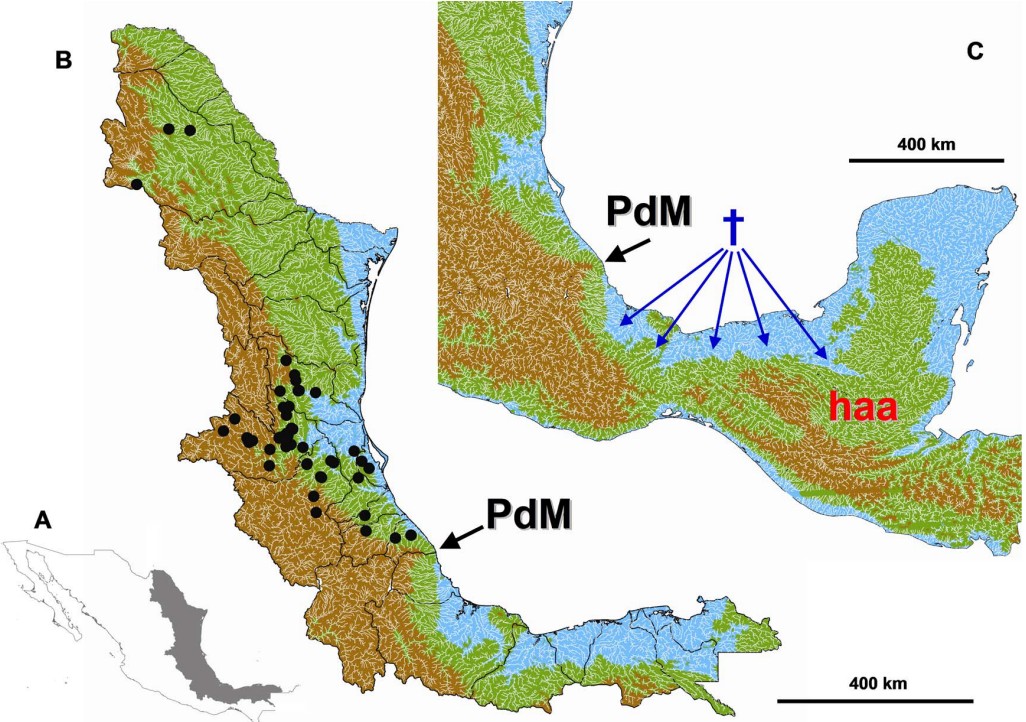

**Figure 1** **Study area.** (A) Study area location in eastern Mexico. (B) Digital elevation model of the Gulf Coast of Mexico depicting the maximum sea level reached during the Miocene and Pliocene. Blue < 60 m a.s.l., Green > 60 < 1000 m a.s.l., Brown > 1000 m a.s.l. The black circles indicate the collection data of the specimens used in this study. (C) The tested geograpical barrier of Punta del Morro (PdM) is indicated with a black arrow, the *Herichthys* ancestral area (haa) is showed in red and reconstructed extinction areas are shown with blue dagger sign and arrows.

CA, USA) with 25-ml reactions containing 1X PCR buffer, 3.0 mM MgCl$_2$, 200 μM of each dNTPs, 0.25 μM of each primer, 40 ng of total DNA, and 1U of Taq DNA polymerase (Invitrogen). PCR conditions were as follows: 95 °C for 5 min followed by 35 cycles of 94 °C for 60 s; an annealing temperature of 60 °C for 70 s; 72 °C for 60 s; and a final extension at 72 °C for 5 min. PCR products were purified with the StrataPrep PCR purification kit (Agilent Technologies, California, USA) and sequenced at Laboratorio de Servicios Genómicos Langebio-Cinvestav Irapuato, Mexico. DNA sequences were edited manually and aligned with Seaview v. 3.2 (*Galtier, Gouy & Gautier, 1996*). Nucleotide coding sequences were also translated into protein sequences to check for possible stop codons or other ambiguities. Sequence data have been deposited in GenBank under the following accession numbers (MK481080–MK481126). Prior to phylogenetic analyses, sequences were collapsed to haplotypes in FaBox (*Villesen, 2007*).

## Phylogenetic methods and molecular clock

Phylogenetic inference was performed using neighbor joining and maximum parsimony (MP) analysis in PAUP* 4b.10 (*Swofford, 2003*) in the first exploratory phase of building the dataset. These analyses were followed by maximum likelihood RaxML analysis with

data partition into 1st+2nd vs. 3 rd position and with a GTR + gamma + I model and bootstrapping as well as a BEAST (*Suchard et al., 2018*) analysis with a relaxed molecular clock model with lognormal distribution of rates and a coalescent model with constant size tree prior. For BEAST analysis four independent runs of 30 millions and sampling each 10,000 generations were performed. After it, runs were verified for convergence with Tracer v.1.10.1 (*Rambaut et al., 2018*) and the trees of the four well-converged runs were combined in LogCombiner v.1.10.1 with a burn-in of 10% and a consensus tree was obtained with TreeAnnotator v.1.8.4, above mentioned analysis well performed in the Cipres server (https://www.phylo.org/).

For divergence time estimation, we used StarBeast 2 (*Ogilvie, Bouckaert & Drummond, 2017*) which compared to BEAST better accounts for species trees vs. gene trees and for intraspecific vs. interspecific events. For the StarBeast analyses we used the traditionally recognized taxa as terminal units in an analytical population size integration model, a Beast named extended substitution model, a birth death model, and either a strict molecular clock or a uncorrelated relaxed molecular clock; four independent runs of 50 millions and sampling each 50,000 generations were performed and the data were analyzed as previously commented for Beast analysis. Finally, the best approach of molecular clock evolution was obtained trough marginal likelihood comparisons following a Nested Sampling approach (*Maturana et al., 2018*).

To calibrate the dating analyses in Starbeast we used five different calibration approaches of the molecular clock in order to assess the timing of the *Herichthys* phylogeny with respect to the main biogeographic hypothesis, i.e., PdM vicariance vs. an earlier colonization prior to formation of PdM. The calibrations are based on fossil and Middle America relevant geological constrains and also use a secondary calibration from one multilocus study that used all available Neotropical cichlid fossils for calibrations (*Musilová et al., 2015*), but most of which are too distant phylogenetically to be applied to our single locus mtDNA phylogeny directly. The same fossils have also been used for calibration in *Tagliacollo et al. (2017)*, who however additionally used the highly questionable constrain of a minimum age of cichlids set at 95 Ma. This additional constrain explains the much older reconstructed dates compared to *Musilová et al. (2015)*. Since the dates are much older due to this additional constrain than *Musilová et al. (2015)* there is no need to include *Tagliacollo et al. (2017)* among our time constraints in testing the much younger PdM. No other fossils and Middle America relevant geological constraints have so far been put forward in publications.

Analysis I follows *Říčan et al. (2013)* and includes one fossil plus two geological calibrations. Calibration nodes: (1) The minimum age of the closest known fossil, *Plesioheros chauliodus* (at 39.9–48.6 Ma; mean 44.25 Ma with SD = 2.7; *Matschiner (2019)*, in their latest review published after our analyses, gives 40–45 Ma as the most probable age of the fossil bed), node all heroine cichlids except *Hypselecara* plus *Hoplarchus*; (2) The split between Cuba and Hispaniola (at 20–25 Ma; mean 22.5 Ma with SD = 1.5), node between Cuban (*Nandopsis tetracanthus*) and Hispaniolan (*Nandopsis haitiensis*) species; and (3) The separation of the Orinoco and Magdalena drainage basins by the final rise

of the Cordillera Oriental (10.1–11.8 Ma; mean 10.95 Ma with SD = 0.6), node between *Caquetaia* sp. cf. *kraussii* and *Caquetaia spectabile*.

Analysis II uses the same two geological calibrations as above but excludes the direct fossil calibration by *Plesioheros chauliodus* owing to its old age and possible influence on the only single mtDNA marker used in the present study (vs. four mtDNA and three nDNA markers in *Říčan et al. (2013)*.

Analysis III on the other hand only uses the fossil *Plesioheros chauliodus* calibration as above to compare its results with just the two geological calibrations.

Analysis IV is based on *Musilová et al. (2015)* who used a wider sampling of fossils for calibrations. We employ here a secondary calibration based on their study, the split of the genus *Caquetaia*, with a mean of 23 Ma, SD = 2.

As described in the introduction the calibration points in analyses I-IV cover the younger range of dated phylogenies that are in conflict with the PdM calibration (i.e., *Concheiro-Pérez et al., 2007*; *Říčan et al., 2013*; *Musilová et al., 2015*). We have thus omitted analyses that would be calibrated from studies that suggest even older dates for cichlid phylogenies (e.g., *Matschiner et al., 2017*; *Tagliacollo et al., 2017*; *López-Fernández et al., 2013*) because these are obviously even more in conflict with the PdM calibration.

Finally, Analysis V employs only the geological PdM calibration and is used to compare the PdM calibration with the other analyses.

## Species delimitation analyses

For species delimitation analyses we have used cytb data since it is the only available dataset with sufficient resolution and with sufficient specimen and locality sampling to be useful for molecular-based species delimitation analyses. We have employed three different delimitation approaches, the General Mixed-Yule Coalescent (GMYC) and the Poisson tree processes (bPTP) that were designed for delimiting species based primarily on single molecular markers, and the coalescent approach implemented in Starbeast. The GMYC model (*Pons et al., 2006*; *Fujisawa & Barraclough, 2013*) is frequently used in empirical studies (*Fontaneto, Boschetti & Ricci, 2008*; *Monaghan et al., 2009*; *Carstens & Dewey, 2010*; *Vuataz et al., 2011*; *Powell, 2012*) and the newer bPTP model (*Zhang et al., 2013*) has been shown to even outperform the GYMC method where distances between species are small. Both methods outperform OTU-picking methods (relying on simple sequence similarity thresholds) and are more robust to cases where the barcoding gap is absent (*Zhang et al., 2013*). The bPTP was run on the RaxML tree and the GMYC analysis was run on the ultrametric tree obtained from BEAST using a single threshold. Both analyses were run at the freely available interface (http://species.h-its.org/) and the GMYC analysis also in the splits library in R (*Ezard, Fujisawa & Barraclough, 2009*). Unlike the two previous methods Starbeast analysis requires that the haplotypes are assigned to *a priori* species. We have used the traditionally recognized species (sensu *Pérez-Miranda et al., 2018*) for this analysis since they are not conflicted by the trees generated by RaxML and Beast (which all gave similar topologies). We have run all three delimitation analyses with the inclusion of only one, the most closely related, outgroup (the genus *Vieja*).

## Biogeographical reconstructions

We use two different hierarchical sets of biogeographical reconstructions using three different molecular data sets in this study. The first biogeographic analysis reconstructs the biogeographic history within the genus *Herichthys* and the ancestral area within the genus based on the mtDNA cytb dataset. As terminal biogeographic units for the analysis we have used *Herichthys* endemism areas (HEAs).

The second set of biogeographic analyses reconstructs the biogeographic history of *Herichthys* within Middle America. For the second biogeographic analysis of *Herichthys* within Middle America we have used two phylogenies, the multilocus yet mtDNA-dominated dated phylogeny of *Říčan et al. (2013)* and the nDNA ddRAD phylogeny of *Říčan et al. (2016)*. The ddRAD phylogeny from *Říčan et al. (2016)*; supplementary material 5 in *Říčan et al. (2016)* is very robust with all basal nodes connecting the major lineages (subtribes) having a posterior probability of 1 and hence with a much better node support than the multilocus mtDNA-dominated phylogenies of *Říčan et al. (2013)* and *Tagliacollo et al. (2017)*. Additionally, there are many significant mito-nuclear conflicts between the mtDNA-dominated phylogenies (*Říčan et al., 2013*; *Tagliacollo et al., 2017*) and the nDNA (ddRAD and exon-based) phylogenies of *Říčan et al. (2016)* and *Ilves, Torti & López-Fernández (2018)* the influence of which on the biogeographical reconstruction we explore in the two phylogenies. The phylogenies of *Říčan et al. (2013)* and *Říčan et al. (2016)* additionally have by far the best taxonomic sampling (both include all but one genus of Middle American and Caribbean cichlids and their most closely related outgroups) and are thus the best suited phylogenies for detailed biogeographical reconstructions. The ddRAD phylogeny of *Říčan et al. (2016)* additionally has better support values than the exon-based phylogeny of *Ilves, Torti & López-Fernández (2018)* that lacks support at several of these nodes and which additionally does not include several important lineages.

For the biogeographical reconstructions within Middle America using the two phylogenies of *Říčan et al. (2013)*; *Říčan et al. (2016)*) we have used cichlid endemic areas (CEAs; sensu *Říčan et al., 2016*) as terminal units, since these are the most fine-scaled units that can be used for the analyses (*Říčan et al., 2016*) and as such provide the most detailed resolution for regional biogeography reconstructions (much more detailed than e.g., the ichthyological provinces used in *Říčan et al., 2013*). The dating of the reconstructed biogeographical events is provided by the phylogeny of *Říčan et al. (2013)*.

Reconstruction of ancestral areas for all nodes in the phylogenetic trees in each analysis (both within *Herichthys* as well as within all of Middle America) was carried out using the event-based Bayesian statistical dispersal-vicariance analysis (S-DIVA; implemented in RASP 2.0; *Yu, Harris & He, 2010*). Distributions of all terminals at the level of the used geographical units were input into S-DIVA. The analyses were carried out using a number of different 'maxareas' options in S-DIVA up to the maximum number of areas in the analysis. If results are the same for all 'maxareas' analyses, then just this one result is reported. If results differ between the 'maxareas' analyses, a summary of all analyses is supplied at the relevant node.

An important point to consider in biogeographical reconstructions is extinctions. In our study of a very dispersal-limited animal group and in the absence of a fossil record, we

reconstructed extinctions in the following manner by taking advantage of the rarely available but in our phylogenies (*Říčan et al., 2013*; *Říčan et al., 2016*; present study) specifically targeted complete species sampling and the use of very fine-scaled biogeographical units (HEAs and CEAs; see above). If in our biogeographical reconstructions at a given node, two or more directly neighboring areas meet, then at this node, there is no need to postulate any extinction and a vicariant event is most likely. In other situations, one or more extinctions (depending on the geographical configuration of the biogeographic units) in the intervening area(s) are postulated.

Our biogeographic analyses are accompanied by DEM simulations of sea-level-caused changes in palaeogeography of northern Middle America based on the sea-level curves of *Haq, Hardenbol & Vail (1987)* and we compare these simulations with paleogeographic maps of past configurations of northern Middle America (https://deeptimemaps.com/). For the DEM analysis, a clip for the study area was extracted from a 30-arc seconds raster from the US Geological Survey (https://geo.nyu.edu) using the raster library 2.3-12 (*Hijmans & Van Etten, 2014*) in R (*R Core Team, 2018*). Afterwards, the raster was imported to ArcMap 10 (*ESRI, 2011*) and the pixels were classified in three categories, <60 m a.s.l. to approximate high sea levels reached during Miocene and Pliocene, up to 1,000 m a.s.l. that is the altitudinal limit that cichlids usually reach and above 1,000 m a.s.l. where we have few rare collections at 1,300 m a.s.l. (Fig. 1).

# RESULTS

## Phylogeny of *Herichthys* based on cytb

The 164 *Herichthys* cyt*b* sequences correspond to 142 haplotypes demonstrating strong population structuring without widespread haplotypes. The phylogenetic relationships of the cyt*b* dataset with the herein extended sampling (Figs. 2–3; Figs. S1–S2) confirmed the results of *Říčan et al. (2016)* and *Pérez-Miranda et al. (2018)*, but also revealed several new points. The two main clades within *Herichthys* are recovered with high robustness as are the relationships between the species with all supraspecific nodes except one strongly supported (Fig. 2; Figs. S1–S2). New results include (1) the non-monophyly of the species *H. tamasopoensis* which was monophyletic in *Říčan et al. (2016)* and *Pérez-Miranda et al. (2018)* and (2) the phylogenetic position of *H. molango*, which has previously not included, and is here found nested within *H. pantostictus*. All species sensu *Pérez-Miranda et al. (2018)* except *H. tamasopoensis* are thus strongly supported by the phylogeny and support values. The species rejected in *Pérez-Miranda et al. (2018)* are thus also rejected here with the addition of *H. molango*.

## Species delimitation analyses within *Herichthys*

Our phylogenetic results strongly support all the species recognized in the review of *Pérez-Miranda et al. (2018)* except *H. tamasopoensis*. Delimitation of the species based solely on the here used mtDNA cyt data is however weaker (Fig. 2). The GMYC and Starbeast delimitation analyses provided more comparable results but failed to separate *H. bartoni* from *H. labridens*, *H. steindachneri* from *H. pame*, and logically *H. tamasopoensis* from *H. carpintis* (since the former is nested as non-monophyletic within the latter). The

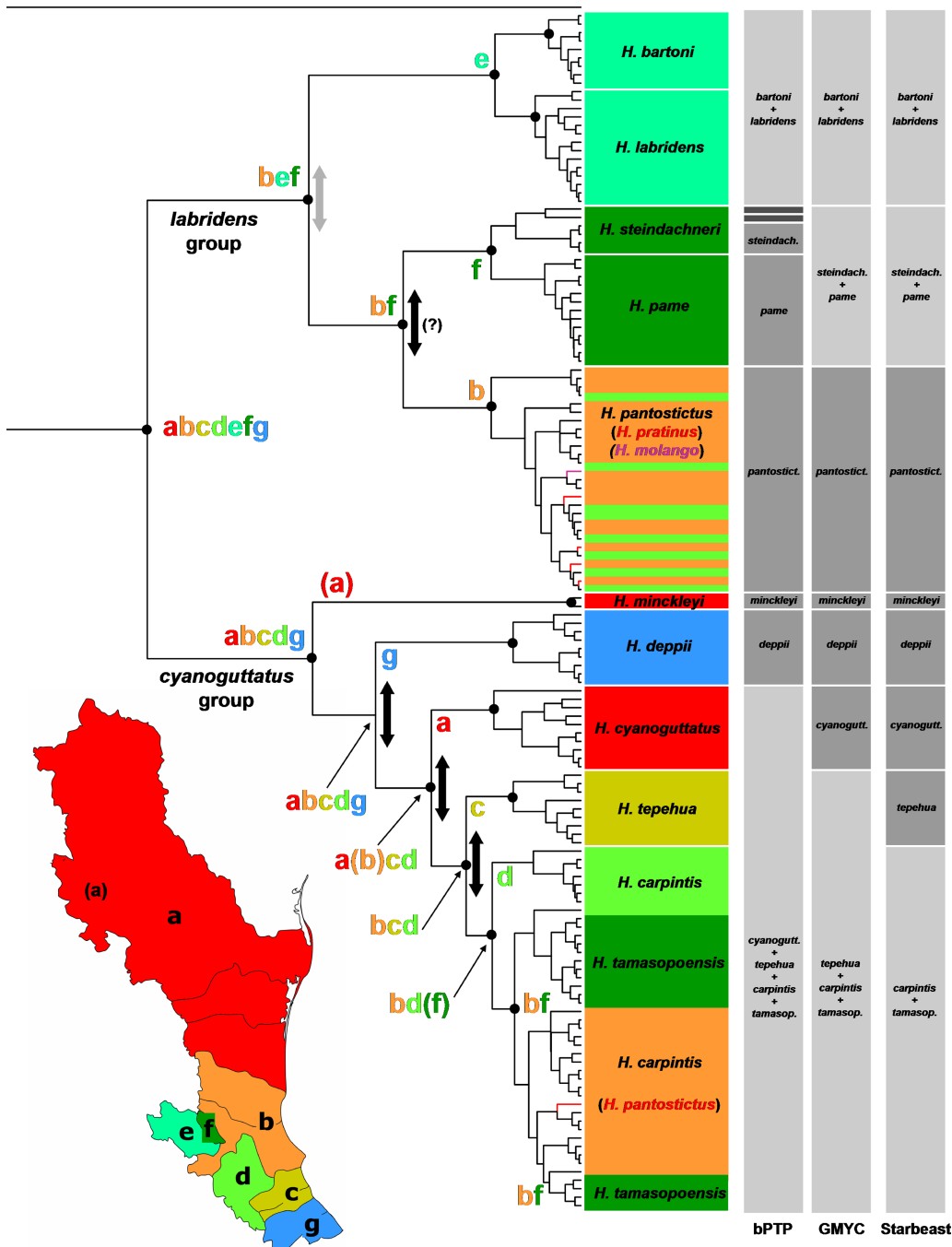

**Figure 2  Phylogeny and biogeography of *Herichthys*.** Phylogeny and biogeography of *Herichthys* based on the cyt*b* haplotype dataset. The topology shown is a BEAST analysis. Species (haplotypes) are color-coded based on distribution areas (*Herichthys* endemic areas; shown in inset map). The endemic areas are optimized on the tree using S-DIVA analysis (ancestral areas at nodes shown by colored letters) which shows a nearly completely vicariant history of *Herichthys*. Vicariant (continued on next page...)

**Figure 2 (…continued)**
events are shown by vertical two-sided arrows; those in black color show vicariant events that correspond to past high sea-level stands (see Fig. 3) and that separate lowland species where sea-level changes could have played a role. The three grey columns to the left of the tree show species delimitation based on bPTP, GMYC and Starbeast analyses. Agreement with present species classification is shown by intermediate grey color, over splitting is shown by dark grey color and failure to delimit recognized species is shown by light grey color. Black dots at nodes show node support above 0.95 in the BEAST analysis (shown only for putative species and deeper nodes; cf. with Figs. S1–S2).

GMYC analysis additionally failed to delimit *H. tepehua* from the previous two species. The remaining species were delimited correctly by both analyses. The bPTP analysis provided somewhat different delimitation since on one hand it only delimited *H. minckleyi* and *H. cyanoguattus* within the *H. cyanoguttatus* group but on the other hand delimited all species in the *H. labridens* group except *H. labridens* from *H. bartoni* and additionally delimited two divergent haplotypes of *H. steindachneri* as distinct species-level groups (Fig. 2).

## Phylogeography and timeframe of speciation events in Herichthys

The biogeographical analysis within the genus *Herichthys* (Fig. 2) reconstructed a wide ancestral area that includes the whole present distribution of the genus. Given the wide reconstructed ancestral node, the majority of *Herichthys* diversification is based on our biogeographic analysis a series of vicariant events (Fig. 2). The separation into the two main clades was nearly completely vicariant (with overlap only in area b, the Pánuco basin; Fig. 2), with the *H. labridens* group after being limited to highlands of the Pánuco basin and *H. cyanoguttatus* mostly to the lowlands of the rest of the area. The basal node in *Herichthys* is dated between 8.9 to 14.4 Ma (Fig. 3).

The *H. labridens* group biogeography is nearly completely vicariant (with sympatry of two species pairs, however, and a later dispersal within *H. pantostictus*). The supraspecific allopatric nodes in the *H. labridens* group are dated between 4.1–6.2 to 5.7–8.9 Ma in analyses I to IV (Fig. 3). The sympatric supraspecific nodes are dated between 0.3–0.5 in the younger pair and 0.8–1.2 Ma in the older pair (Fig. 3).

The *H. cyanoguttatus* group biogeography is completely vicariant with the first two species to diverge being the northernmost (*H. minckleyi*) and southernmost (*H. deppii*), followed later by more tightly spaced vicariant events in the central area, that again included a northern (*H. cyanoguttatus*) and southern species (*H. tepehua*) diverging from the centrally distributed species (*H. carpintis*, which in our analyses contains the parapatric *H. tamasopoensis*). The allopatric supraspecific nodes in the *H. cyanoguttatus* group are dated between (except the *H. carpintis/H. tamasopoensis* node) 1.6–2.4 to 4.4–7.0 Ma (Fig. 3). The speciation events in both species groups thus fall into the same time window. Our biogeographic reconstructions suggesting vicariance as the predominant speciation mode for the allopatric species and the reconstructed dates (analyses I–IV) for these events in the lowland species correspond with high sea-levels that existed during the Miocene/Pliocene boundary (Fig. 3).

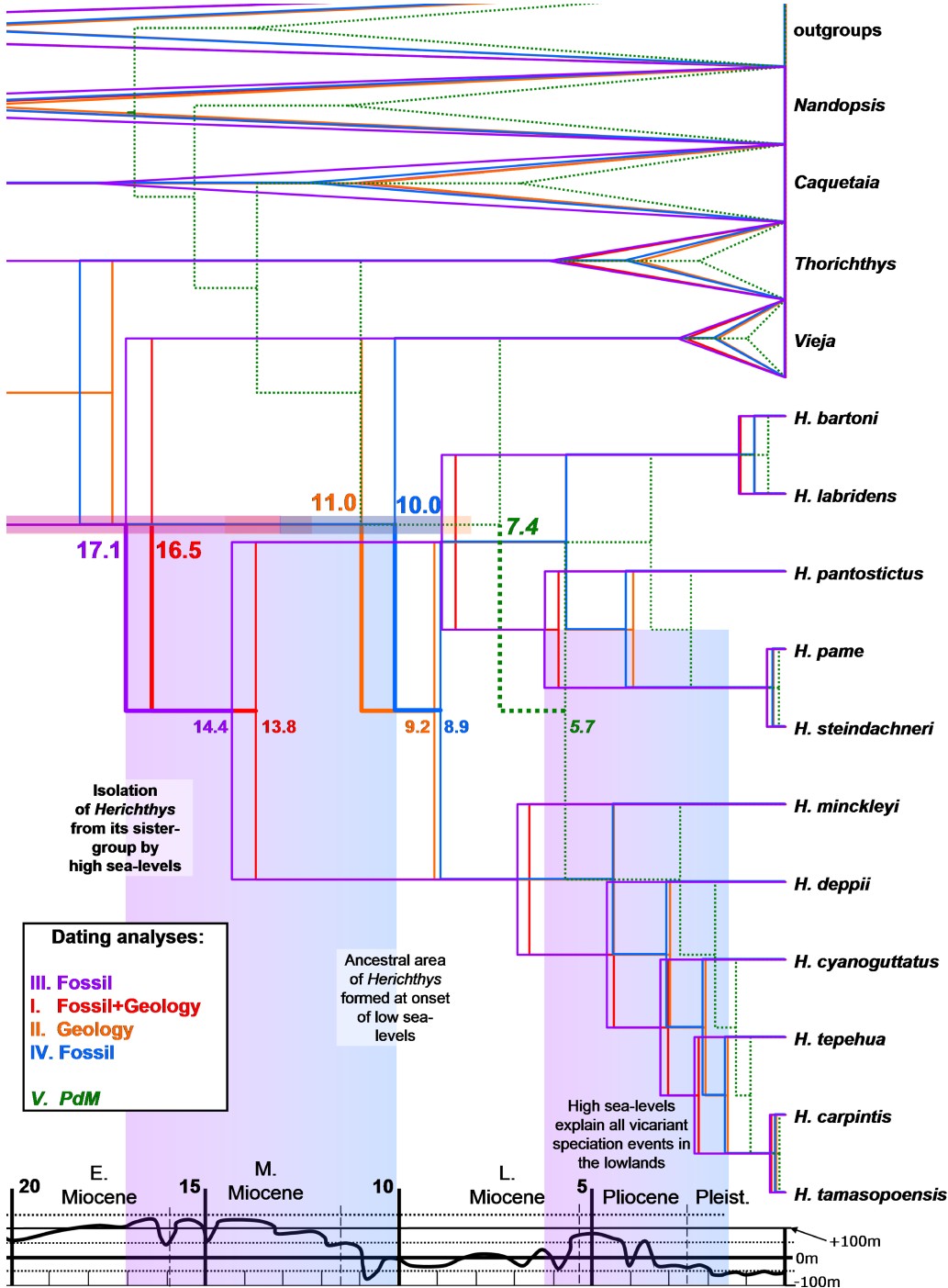

**Figure 3 Dated phylogenies of *Herichthys* from Starbeast.** Dated phylogenies of *Herichthys* from Starbeast based on the cytb dataset using the five calibrating analyses. Median dates for divergence of *Herichthys* from its closest sister-group (upper numbers; with 95% HPD confidence intervals) and for the basal node of *Herichthys* (lower numbers) are shown. The colored columns show correspondence of *Herichthys* divergence (left column) and of *Herichthys* allopatric speciation events in the lowlands (right column) with high sea-levels. The divergence of *Herichthys* from its most closely related clades is thus a result of past high sea-level stands that occurred during the middle to late Miocene as shown by the good correspondence.

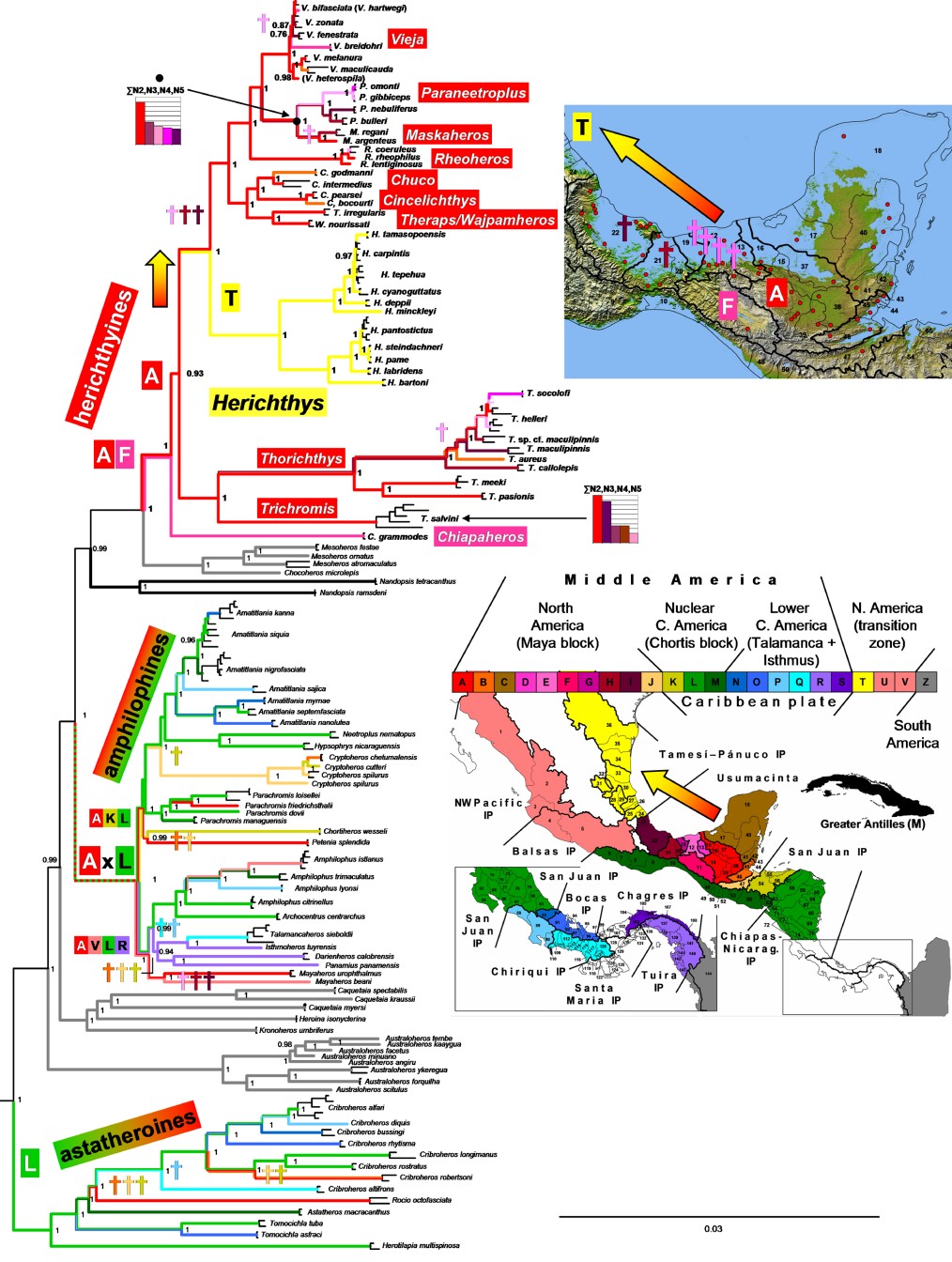

**Figure 4  Biogeographical reconstruction of the Middle American cichlids.** Biogeographical reconstruction of the Middle American cichlids based on the ddRAD phylogenetic hypothesis of *Říčan et al. (2016)*. Reconstruction for nodes above the three Middle American clades is omitted from the figure. Reconstructed extinction events are shown by dagger signs that are color-coded in correspondence to the area where each extinction event occurred. Note that *Herichthys* is not separated from its closest relatives by a vicariant event but by a node with a large zone of reconstructed extinctions (shown in the upper inset map) following colonization (shown by the red to yellow arrow). For the same biogeographic analysis using the mtDNA-dominated multilocus dataset and for dating of the events including the identified extinctions in both analyses, see Fig. S3.

## Dating of Herichthys at PdM

Our dating analyses of the *Herichthys* cytb phylogeography (Fig. 3) reject the postulated PdM vicariance and favor older dispersal prior to the formation of the PdM using either a strict or a relaxed molecular clock. Due that similar results were obtained with both approaches, we only present the divergence dates of the strict molecular clock which obtained a higher likelihood value in the five calibrations (Table S2). All analyses that exclude the PdM calibration (analyses I-IV) reconstruct the age of *Herichthys* diversification to be between two and three times older than the PdM calibration in Starbeast analyses (10.0 to 17.1 Ma) (Fig. 3) vs. 5 to 7.5 Ma of PdM. The PdM calibration analysis (calibrated with 7.5 Ma; Fig. 3) also reconstructs a very young overall cichlid diversification time frame.

## Biogeography of cichlids at PdM

The reconstructed dates of divergence of *Herichthys* from its sister groups 10.0 to 17.1 Ma correspond to high sea-levels during the early and middle Miocene (Fig. 3). Further correspondence with global sea-levels is seen in the reconstructed date of the basal node of *Herichthys*, which falls into the time period when the sea levels experienced a significant drop from more than 100 m above present sea levels to close to present-day sea levels during the late Miocene (*Haq, Hardenbol & Vail, 1987*; Fig. 3, Fig. S3). The wide reconstructed ancestral area within the genus Figs. 2 also corresponds with the sea-level changes since prior to the late Miocene sea level drop a continuous ancestral area of the genus would not have been possible (Figs. 1 and 3).

The Miocene/Pliocene divergences between the lowland species of *Herichthys*, likely caused by the high sea-levels (Fig. 3), are based on our biogeographical reconstructions and datings of the whole Middle American cichlids accompanied by contemporaneous extinctions among the closely related allopatric lowland genera *Vieja* and *Maskaheros* (Fig. 4 and Fig. S3).

The early to middle Miocene reconstructed date of divergence of *Herichthys* from its sister groups (10.0 –17.1 Ma in the Starbeast analyses I-IV, Fig. 3), corresponding and thus likely also caused by the high sea-levels (Fig. 3), is also accompanied by reconstructed extinctions (Fig. 4; Fig. S3), in this case in the whole intervening area between the *Herichthys* ancestral area (Fig. 2) and the ancestral area of the whole herichthyine clade (Fig. 4; Fig. S3). All the reconstructed extinction events are localized into the lowland area between the two reconstructed ancestral areas strengthening the cause by the high sea-levels. Since the reconstructed extinctions exhibit very robust correspondence with increased sea levels, and since our dating analyses are in conflict with the PdM as a vicariant event, the isolation of the *Herichthys* ancestor was thus more likely achieved by high sea levels than by vicariance caused by the formation of the PdM. The differentiation of *Herichthys* from the remainder of its clade was thus not a vicariant event, but a dispersal event followed by isolation caused by local extinctions.

## DISCUSSION

### Species diversity and Biogeography of the genus *Herichthys*

Species diversity in *Herichthys* is complex (reviewed by *Pérez-Miranda et al., 2018*) and the expanded sampling in the present study strongly supports ten species. The herein employed species delimitation methods did not provide converging or inspiring results. The results of all three analyses (GMYC, bPTP and Starbeast) all suggest a more conservative classification delimiting in agreement only eight species. The recently diverged sympatric sister-species that are highly divergent morphologically (*Pérez-Miranda et al., 2018*) have not been recovered by the delimitation methods, even though they form in all here employed phylogenetic analyses reciprocally monophyletic and strongly supported species.

Two recently described species (*H. pratinus* and *H. molango*) and one traditionally recognized species (*H. tamasopoensis*) are not supported by any of the phylogenetic analyses nor delimitation analyses of the cytb dataset (Fig. 2) because they are not monophyletic and their non-monophyly within two other species (*H. pantostictus* and *H. carpintis*; Figs. S1–S2).

The interesting (or problematic) result of the cytb species delimitation analyses are the deep clades within *H. steindachneri* as the species is a relatively recently diverged sympatric sister species of *H. pame*, and it was hypothesized from the larger distribution and ecomorphology of *H. pame* that this species is closer to the ancestor of the species pair (*Artigas-Azas, 2008*; *Pérez-Miranda et al., 2018*).

The limited resolution of the molecular species delimitation analyses coupled with only modest morphological differentiation, especially within the two species groups well demonstrates the difficulty of species classification in the genus. The employed single locus species delimitation, such as GMYC and bPTP, are fast protocols that allow identifying putative species; however, they have several limitations (*Amado, Farias & Hrbek, 2011*; *Colatreli et al., 2012*; *Farias & Hrbek, 2008*; *Willis, 2017*; *Carvalho et al., 2018*; *Machado et al., 2018*). Further studies including independent genomic data with a robust locality sampling are thus necessary to provide hypotheses of species boundaries (*Zhang et al., 2011*). On the other hand, Starbeast is a method designed to estimate both gene tree and species tree under a coalescent theory from a large number of markers that avoid the caveats of concatenation of markers (*Ogilvie, Bouckaert & Drummond, 2017*). However, it requires the definition of "a priori" taxon labels that could be useful to synonymize species but not to postulate putative cryptic species.

Based on results of molecular clock dating, *Herichthys* colonized its present distribution area significantly prior to the suggested vicariance by PdM 5 to 7.5 Ma (*Hulsey et al., 2004*; *Hulsey, Hollingsworth & Fordyce, 2010*). The colonization of the present area occupied by *Herichthys* occurred based on our analyses prior to 10.0–17.1 Ma (based on Starbeast analyses I–IV), i.e., the dated split between *Herichthys* and its sister-groups. The proposed divergence of this genus (and many other groups where this calibration has been used; see Introduction and below) at the PdM as a vicariance event is thus not compatible with other dating constraints as demonstrated in the present study, the PdM calibration being a clear outlier among the dating constraints.

The use of the PdM calibration in order to test events across the PdM is additionally clearly a product of circular reasoning (*Kodandaramaiah, 2011*; *Ho, Pruett & Lin, 2016*). Such a situation has to be avoided, either by excluding the vicariant event in question from the calibration or by using it together with other calibrations (*Kodandaramaiah, 2011*; *Reznick et al., 2017*) as has been done in this study. Previous studies that utilized this circular reasoning in dating of divergences include the cichlids and *Herichthys* in particular (*Hulsey et al., 2004*; *Hulsey, Hollingsworth & Fordyce, 2010*), and also non-cichlid genera such as *Astyanax*, *Pseudoxiphophorus*, and *Xiphophorus* (*Ornelas-García, Domínguez-Domínguez & Doadrio, 2008*; *Agorreta et al., 2013*; *Culumber & Tobler, 2016*). *Palacios et al. (2016)* found an estimated age of 5.28 Ma for the subgenus *Mollienesia*, using a universal mitochondrial mutation rate that coincided with the formation of PdM, but with the use of a secondary calibration point of *Ho, Pruett & Lin (2016)* that is based on a wide fossil record (*Betancur-R et al., 2013*), the time estimates were reduced drastically to 1.28 Ma and therefore postdating the vicariant hypothesis for the species north and south of PdM.

The reconstructed biogeography of the separation of *Herichthys* from its closest relatives provides an even stronger case (as opposed to just dating) for colonization prior to PdM instead of vicariance at PdM. Reconstruction of ancestral areas in both the nuclear ddRAD topology (Fig. 4) as well as mtDNA-dominated topology (Fig. S3) identifies the same set of extinctions in the intervening area between *Herichthys* and its closest relatives. The dating of the reconstructed extinctions (Fig. S3) clearly demonstrate that isolation of *Herichthys* from its closest relatives is older than the formation of the PdM (10 –14 Ma vs. 7.5 to 5 Ma in the original dating of *Říčan et al., 2013*; see Fig. S3; 10–17. 1 Ma in analyses in this study; Fig. 3). These reconstructed extinctions additionally exhibit a very strong correspondence with increased sea levels that existed during the period of the divergence in early and middle Miocene (24–12 Ma; Figs. 1 and 3; Fig. S3). The isolation of *Herichthys* from its closest relatives was thus not a vicariant event at the PdM but a dispersal event followed by isolation through extinctions probably caused by high sea levels.

## Implications for the Biogeography of Middle America and the Caribbean

The geological literature offers several dating constraints for colonization and diversification of cichlids in Middle America and the Caribbean. Except for one, the PdM, none were so far used in any publications focusing on cichlid colonization and diversification in Middle America and it is thus timely to introduce them here and compare them with the results of this study and published dated phylogenies of the Middle American cichlids and cichlids in general. The geological constraints for cichlid evolution in Middle America and the Caribbean are of great importance to the debate regarding cichlid dating, in general, as Middle America and the Caribbean is the only area of cichlid distribution where the group occurs on present or former relatively small islands within former island chains (*Říčan et al., 2013*), while most other cichlids are continental (even Madagascar is much larger than any of the islands in past and present configurations of Middle America and the Caribbean). Only a handful of cichlid species among the thousands of species is known from marine or brackish conditions (*Kullander, 1983*; *Murray, 2001*; *Říčan et al., 2016*) and the degree

of capability of marine crossings remains debated, but an island chain setting clearly does provide better constraints on cichlid biogeography than do continental settings.

The oldest geological evidence for continuity in Caribbean land environments is from 37 Ma (*Iturralde-Vinent & MacPhee, 1999*; *Iturralde-Vinent, 2004a*; *Iturralde-Vinent, 2004b*; *Macphee & Iturralde-Vinent, 2005*). This means that older terrestrial (freshwater) habitats that would have remained subaerial are not known at the moment. Therefore, all postulated colonizations of the Greater Antilles older than this date (e.g., *Tagliacollo et al., 2017* at 45–50 Ma for cichlids and 56 Ma for poeciliids; *López-Fernández et al., 2013* at 55 Ma for cichlids; and, marginally, *Matschiner et al., 2017* between 44 and 31 Ma) do not explain where and how these fishes have survived in the Antilles. During these old times, there was virtually no land in the Antilles—they were almost completely under the sea. Similarly, before ca 35 Ma (all through the Cretaceous from 75–70 Ma), most of the Maya block was also under sea (*Iturralde-Vinent, 2004a*; *Iturralde-Vinent, 2004b*). Continuous land, however, existed in the area of the present diversity center of cichlids (*Říčan et al., 2016*) in the south of the Maya block. Colonization dates older than 37 Ma would thus first need to reach Middle America (the Maya and Chortis blocks) and only later from there colonize the Greater Antilles. The Chortis block continued its movement toward the east and south from west of present-day SE Mexico and in the Lower Oligocene, 35 Ma, united with the Maya block along the presently still active sutures, defining the present territory of Guatemala. Before this date, the Chortis and Maya blocks were not in contact and, additionally, large portions of both were covered by shallow seas (from *Iturralde-Vinent, 2006a*; *Iturralde-Vinent, 2006b*).

The Chortis block underwent massive volcanic activity that created the middle Miocene ignimbrite province (the High Volcanic Plateaus, HVP) between 20–14 Ma (*Rogers, Kárason & Van der Hilst, 2002*; *Rogers, Mann & Emmet, 2007*; *Jordan et al., 2006*; *Molina-Garza et al., 2012*). This volcanic activity was directly linked with the docking of Lower Central America (LCA) in the form of the Miocene Volcanic Arc (MVA) at 22 Ma and its subduction under the Chortis block (*Coates & Obando, 1996*; *Coates, 1997*; *Coates et al., 2004*; *Kirby & MacFadden, 2005*; *Kirby, Jones & MacFadden, 2008*). At the same time also began the formation of the San Juan basin between the southern terminus of the Chortis block and LCA. Interestingly, cichlid fishes, based on biogeographic analyses (Fig. 4 and Fig. S3), underwent significant extinctions in all areas of the HVP and most neighboring northern areas on the Chortis block and, importantly, virtually nowhere else in Central America. These extinctions are independently dated between 24–13 Ma (following the dating analysis by *Říčan et al., 2013*; Fig. S3). The extinctions happened not only in highland areas of the HVP, but also in the low-lying areas of the Chortis block that lie outside major concentrations of the ash falls. The datings of the extinctions (24–13 Ma) in the highlands coincides with the HVP (20–14 Ma), while extinctions in the lowlands coincide as well with HVP as well as with high sea levels during the Late Oligocene to Middle Miocene (23–14 Ma). The coincidence is striking and we are not aware of any other events that would explain these extinctions that produced the depauperate cichlid faunas of northern Nuclear Central America (Honduras, El Salvador, northern Nicaragua) and
the concentration of all lineages in the San Juan basin, the youngest area of Nuclear Central America (*Říčan et al., 2013*; *Říčan et al., 2016*).

The emergence of LCA in the form of the MVA (22 Ma) posits the oldest possible date for colonization of LCA and of the San Juan basin because before that, these areas were submerged below the sea. The cichlid colonizations of LCA, as independently dated by *Říčan et al. (2013)*; Fig. S3, started at the latest by 18 Ma, and all lineages were established in LCA at the latest by 9 Ma.

The novel dating constraints for cichlid diversification in Middle America can thus be summarized as follows: (1) Maximum and minimum age for colonization of Middle America through the Greater Antilles 37–33 Ma. The dating by *Říčan et al. (2013)* is thus 6 Ma below the maximum age (Fig. S3); (2) Maximum age for colonization of LCA is 22 Ma. The dating by *Říčan et al. (2013)* is thus 4 Ma below the maximum age; (3) Age of HVP 20–14 Ma and dated cichlid extinctions in HVP 24–13 Ma. The dating by *Říčan et al. (2013)* is thus for this node at its maximum age; (4) The extinctions of cichlids in low-lying areas were likely influenced by the high sea levels of the Late Oligocene to Middle Miocene (23–14 Ma) and have a terminal age of 12.5 Ma on the Chortis block and 10.5 on the Maya block. The dating by *Říčan et al. (2013)* is thus only 1.5–3.5 Ma younger than maximum age; (5) The extinctions of cichlids in low-lying areas of the Maya block likely caused by the high sea levels during the early Pliocene (5.5–4.5 Ma) are in *Říčan et al. (2013)*; Fig. S3 dated exactly within this timeframe and thus the cichlid dating at these nodes cannot be any older or younger.

The aforementioned geological constraints on dating of cichlid phylogenies in Middle America postulate that cichlid evolution in the area cannot be significantly older (and definitely not younger) than proposed in the dated framework of *Říčan et al. (2013)* and that the five constraints are coherent among each other. All of these five novel geological constraints, as well as the constraints used in the present study, in *Říčan et al. (2013)*, in *Musilová et al. (2015)*, in *Tagliacollo et al. (2017)*, or in *Matschiner et al. (2017)* are on the other hand in conflict with the PdM constraint, since all give much older divergences at the PdM than are compatible with vicariance caused by the PdM. The PdM calibration on the other hand results in one of the youngest dates for cichlid evolution (this study; *Hulsey et al., 2004*; *Hulsey, Hollingsworth & Fordyce, 2010*), based on this study even marginally younger (at the comparable node of *Hypselecara* divergence from its sister-group) than in *Friedman et al. (2013)*, so far the youngest and clearly outlying study of cichlid divergence dates (*Matschiner, 2019*). In summary, the five novel constraints examined here in the discussion of Middle American palaeogeography as bounds for colonization and diversification of cichlids in Middle America and the Greater Antilles suggest that the cichlid diversification in Middle America can be, at the most, 6 Ma, 4 Ma, and 1.5–3.5 Ma older (in the respective succession of nodes from oldest to youngest) than the dating frame reconstructed by *Říčan et al. (2013)*, while the two youngest constrained nodes are at their maximum age. This demonstrates that any cichlid datings significantly older (e.g., *López-Fernández et al., 2013*) and all other studies that used Gondwana vicariance as the sole calibration constraint; as well as (*Tagliacollo et al., 2017*); and marginally (*Matschiner et al., 2017*) or younger (*Hulsey et al., 2004*; *Hulsey, Hollingsworth & Fordyce, 2010*; *Friedman et al., 2013*) than the bounds

presented here have to be taken as highly questionable from the point of view of Middle American paleogeography and cichlid biogeography unless we allow the option that cichlid biogeography is completely independent from ecological and geological constraints.

## CONCLUSIONS

The results of this study suggest that in opposition to the majority of previously published literature regarding freshwater fish divergence across the Punta del Morro (PdM) the PdM has in the here studied genus *Herichthys* not acted as a vicariant event (at 7.5 to 5 Ma), but instead the ancestor of *Herichthys* colonized its ancestral distribution area to the north of it through a dispersal event prior to 10 to 17 Ma and that the separation of *Herichthys* from its closest clades was due to extinctions in the intervening area caused by high sea levels during the early to middle Miocene, not the result of the uplift of Punta del Morro between 5 to 7.5 Ma. The old divergence of *Herichthys* is also supported by additional novel constraints from Middle American geology and palaeogeography introduced in this study that suggest that compared to the dated framework of cichlid evolution in Middle America of *Říčan et al. (2013)* the actual evolution of the group in Middle America can be, at the most 1.5 to 6 Ma older and definitely not younger. This demonstrates that any published cichlid datings significantly older or younger than the bounds presented here have to be taken as highly questionable from the point of view of Middle American paleogeography and cichlid biogeography unless we allow the option that cichlid biogeography is completely independent from ecological and geological constraints.

## ACKNOWLEDGEMENTS

We thanks to Juan Carlos Fontanelli from Balneario cascadas de Tamasopo and the staff of Rancho el Jobo for the facilities provided for specimen collection. We also thanks to the associate editor Thomas Hrbek and two anonymous reviewers for their helpful comments to improve the manuscript.

### Funding

These work was funded by SIP projects number 20180849 and 20190100. There was no additional external funding received for this study. The funders had no role in study design, data collection and analysis, decision to publish, or preparation of the manuscript.

### Grant Disclosures

The following grant information was disclosed by the authors:
SIP: 20180849, 20190100.

### Competing Interests

The authors declare there are no competing interests.

## Author Contributions

- Fabian Pérez-Miranda and Benjamín López performed the experiments, analyzed the data, prepared figures and/or tables, authored or reviewed drafts of the paper, and approved the final draft.
- Omar Mejia and Oldřich Říčan conceived and designed the experiments, analyzed the data, prepared figures and/or tables, authored or reviewed drafts of the paper, and approved the final draft.

## Animal Ethics

The following information was supplied relating to ethical approvals (i.e., approving body and any reference numbers):

Animals were euthanized according to the specifications described in the Mexican law NOM-033-SAG/ZOO-2014.

## DNA Deposition

The following information was supplied regarding the deposition of DNA sequences:

The sequences generated in this study are available at GenBank: MK481080–MK481126.

## Data Availability

The raw measurements are available in the Supplemental Files.

## Supplemental Information

Supplemental information for this article can be found online at http://dx.doi.org/10.7717/peerj.8818#supplemental-information.

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
