# Peer review of "Molecular clocks, biogeography and species diversity in Herichthys with evaluation of the role of Punta del Morro as a vicariant brake along the Mexican Transition Zone in the context of local and global time frame of cichlid diversification"

_PeerJ, doi:10.7717/peerj.8818_

## Round 0.1 · original submission · Major Revisions

Dear Authors,

I received two thorough reviews of your MS. Both reviewers agree that a major revision is needed, but one of the reviewers is much more positive in their evaluation than other.
Given their thorough reviews, I will not get into an exhaustive review, but there are a couple of points that I would like to point out.

Introduction could be reduced. In many sections it is quite repetitive.
Result has many components that belong to the discussion.

1) The lack of phylogenetic support for species relationships in single locus species discovery/delimitation analyses is not much of an issue since the objective is to distinguish between intra and inter-specific species divergences, and this is not influenced by lack of phylogenetic support or resolution at deep divergences. I would make this clear in your MS.
2) Discordance between mtDNA and ddRAD phylogenies. First, this may be just to differences in which individuals were included in which dataset. Discarding these cases, I would test if there is discordance in species discovery/delimitation using both datasets. If there is discordance, then this discordance has a reason, and this in itself could be discussed. But more importantly a signal of allopatric divergence in the mtDNA but not in the ddRAD data still points to an allopatric divergence of lineages but this barrier was transient or porous enough to prevent or secondarily erase divergence in the nuclear genome. But there still is a signal of this event.
3) Having said the above, the link between the mtDNA and ddRAD data is not clear. Delimitation analyses are done on the mtDNA data, but then independent of the results, the ddRAD phylogeny is used for biogeographic analyses. There needs to be a link.
4) I agree that phylogenetic and dating analyses need to be redone. In particular you need to eliminate the PdM calibration point, and you should use STARBEAST species tree analyses to be able to account for both intra and inter-specific processes within your data. Or alternatively generate your phylogenies, delimit species/lineages, choose a representative individual, and then run the dated analyses and run dated analyses on the ddRAD dataset as well sampled at species level based on the mtDNA delimitation. Also it is not clear why the diversity of calibrations were used. Justify (make more explicit) the use of the different schemes.
5) Justify better the use of the cichlid areas of endemism.

General
I find it quite strange that the GMYC analysis delimited many fewer taxa than PTP, and even delimited as one taxa that are clearly divergent. Please check this is not some kind of an artifact. Normally, GMYC tends to oversplit, and PTP is a more conservative method which again makes me think something is not correct here.
Finally, the way I read your MS is that it has two principal goals and sections. First is an expanded analysis of species diversity, and then the second is the test of the PdM as an agent of vicariance. For this second part you do not need to have a full species (and all the lineages discovered) sampling in the ddRAD data. As the MS is written, it is not clear what the links between these two datasets are, and why one is being used for some analyses and another for other analyses.

L204-205 use full species names
L292 PTP analyses should not be run on ultrametric trees. For PTP analyses branch lengths are meant reflect substitutions rather than time, while in BMYC branch lengths are meant to reflect time (ultra metric trees) rather than substitutions
L332 “raster” package should be in small caps
L354 on which phylogeny (MrBayes or BEAST) was the bPTP analyses based?
L561 it is too strong to state that “cichlids hence must be considered strictly freshwater fishes”. But I understand why you are making this statement. However, I think the best evidence that the separation of Hispaniolan and Cuban cichlids is a vicariant event driven by the separation of the islands is that this divergence time is also recovered from other independence geological calibration points. And having this independent evidence, then I think you can safely use the Hispaniola/Cuba separation in your analyses.
L638-652 you have not addressed these issues in your MS, and so remove this paragraph from the discussion

I look forward to seeing your revised MS in the near future.
Sincerely,

Tomas Hrbek

·

Basic reporting

English requires a revision by a native speaker.

The authors did not clearly give credit for some of the ideas presented in their manuscript.

The article is poorly formatted and is not structured, the figures are poorly formatted, and not clearly described in the results section.

The data is shared, but their methods are not clearly described.

The article fails to clearly state the major advances with the previous studies published by the authors.

Experimental design

I found this a problem since they did not clearly state how this MS is original compared to the previous studies published by the coauthors.

The research question is defined but the tools used to resolved the question are questionable.

Some aspects related to giving credit to other authors ideas, as well as rigorously describe the background to their project is necessary. Finally, there are some major methodological flaws that require attention.

They need to better describe the methods.

Validity of the findings

Not a clear impact nor novelty on the study.

The study required to be improved in their analyses.

Several sections are very speculative and not clearly identified.

Additional comments

The authors used a combination of cytb and ddRADseq data, to carried out a species delimitation analysis and biogeographic analysis of the Herichthys genus. In particular, authors used the ddrad phylogenetic reconstruction to test the role of the Punta de Morro (PdM), as a vicariant event in the evolutionary history of the genus. While the Herichthys genus represents a very interesting model system to further explore species delimitation methods, I found the document with several organization problems as well as several methodological flaws (i.e. one maker instead of the genomic data in the species delimitation), that makes very difficult to understand the major advances in comparison with previous studies already published by the authors, then I did not found the MS particularly groundbreaking. Moreover, most of the findings were largely descriptive and the comparison with other studies at some sections were either vague or speculative. Finally, as I already mentioned one of the major flaws correspond to the use of only one marker in the species delimitation in comparison with the use of genomic data, mainly considering that the authors claim that the best phylogeny was recovered with the Radseq data. Beyond that, there are several studies that have pointed out the incongruencies between the gene trees and the species trees, mainly considering the authors' previous studies where they have suggested bias in the use of a single marker. Thus, due to the major issues I previously describe, I cannot recommend the MS to be published at PeerJ.

Here I include some major concerns, that I hope that can help the authors to improve their manuscript.

In general, I found the MS hard to follow, mainly because the document is not well organized, mixing the results with the discussion and without an order according to the methods section, making very complicated to understand the document, thus I strongly recommend the authors to carefully review and organize the document.

Additionally, through the MS the authors claim that the phylogeny of the Middle American cichlids is complete, “L117 The phylogeny of Middle American cichlids is now completely known based on complete species-level sampling and genomic phylogenies (Říčan et al. 2016)”, I found two problems with this statement, the first if that a phylogenetic hypothesis could be robust, but “completely known”, and moreover the authors fail to clearly stand for the advances of the document if everything is already known.

I found concerning that, in general, the authors did not give the credit to some of the ideas presented in their MS, but particularly important are the previous studies where the PdM has been suggested as a barrier for freshwater species (i.e. Contreras-Balderas, S., Obregon-Barboza, H., & Lozano-Vilano, M. L. (1996). Punta el Morro, an interesting barrier for distributional patterns of continental fishes in North and central Veracruz, México.). I encourage the authors to be more careful giving the credit to ideas suggested as theirs.

Either in the introduction as well as in the discussion, some general mechanisms or processes are not properly introduced, or is not clear how the authors could provide those analogies, making difficult to understand those ideas (i.e. A clear distinction between the central American cichlids and the North American cichlids is the conspicuous differences in the habitats, the first corresponds to the volcanic crater lakes, while the second ones correspond to a polymorphic groups which divergence have been clearly pointed in the plasticity directions.).

Methods
One major concern is that the authors did not follow an order between the methods, results, and discussion. Moreover, some sections of the methods are not clearly described, and a clear example of this was the poor description of the molecular clock used in the study, particularly, since the description did was in accordance with the result and figures presented. I strongly recommend the authors to clearly state how the run the molecular clock analyses, particularly describing if they partitioned the data matrix, which was the substitution model used for each partition, and how they apply the molecular clock to the ddRadSeq data.

By the other hand, the use of one mitochondrial marker to the species delimitation analyses was very confusing, since the authors used the ddRadseq data for the Biogeographic analysis carried out in this document. Moreover, in the document the authors claimed that the ddRad topology was the most robust and detailed, thus it seems contradictory to keep the less resolved and detailed topology in the delimitation analyses.

Additionally, there is large evidence that the use of mitochondrial DNA exclusively in the species delimitation could have some troubles (i.e. Brower, A. V. (2006). Problems with DNA barcodes for species delimitation:‘ten species’ of Astraptes fulgerator reassessed (Lepidoptera: Hesperiidae). Systematics and Biodiversity, 4(2), 127-132.) Maddison, W. P. (1997). Gene trees in species trees. Systematic biology, 46(3), 523-536.; Pamilo, P., & Nei, M. (1988). Relationships between gene trees and species trees. Molecular biology and evolution, 5(5), 568-583.)).

By another hand, at the biogeographic analyses, I found problematic the definition of the endemism areas, since in some cases they correspond to the same basin, thus it is not clearly stated how they defined those regions. For example, the authors divide the upper and lower Panuco basin under which criteria. Moreover, I consider very important to clearly stand which are the improvements-differences to the previous studies published in these biogeographic scenarios, that makes the current study and original contribution. Finally, authors should better describe the fundaments for some claims related to the biogeographic patterns, since some of them are clearly contradictory to previous studies (i.e. Rosen, D. E. (1975). A vicariance model of Caribbean biogeography. Systematic Biology, 24(4), 431-464.)

Results

In my opinion the results section is the less organized in the document, particularly because the authors did not state the major differences between their phylogeny (obtained with the cytb) in contrast to the ddRadseq topology, instead, apparently they used the ddRAdseq topology to describe the species recovered in both delimitation methods. I found this as the major flaw of the document since it is not clear which is the major advance compared to previous studies, and moreover, why did the authors use the ddRadseq three to described their valid species, if the species delimitation was carried out with the cytb data. Thus, the species delimitation results seem unnecessary ambiguous in terms of which are the major differences with the previous published data.

Species delimitation results are poorly described, particularly since the authors claim some differences between the two methods used (e.g. 476-481), but those major differences were not accurately described, making confusing to understand which figure 2 or 3, the one describing the major differences across the methods. Additionally, authors did not accurately describe why they used the Radseq topology to present the valid species. Moreover, Figure 2 was not accurately described since on it least three alternative taxonomic proposals appeared but they are not clearly described in the document.

It is not clear how the authors could suggest that they apply a molecular clock using the ddRadseq data (L387-393), thus this is very important, since they need to clearly describe how they construct apply the molecular clock to the Radseq data, this is particularly important considering that it is not possible to assign a mutation rate to SNPs data.

In general, the results section resembles more a discussion section, were repeatedly the authors contrast their results with other studies, or even give some explanations to the patterns found.

Discussion
Similarly to the results section, the species delimitation and the biogeographic scenarios are commonly mixed, making very complicated for an unfamiliar reader understand the scenarios described, and the hypotheses tested. Moreover, most of the major conclusions in the document make reference to the previous results presented in Říčan et al. 2016, making very difficult to understand the relevance of the present study.

Similarly to the introduction, at the discussion authors commonly introduce some points that have not been properly defined, as “the ecomorphological polymorphism within an species group” (ie. L487), or “deep clades” (ie. L488), this makes even more vague the arguments related to the diversification of the group, making most of those sections speculative.

The authors suggest that “Further studies including independent genomic data with a robust locality sampling are thus necessary to provide hypotheses of species boundaries (Zhang et al. 2011).” could help in the species delimitation hypothesis, however, the authors did not use the genomic data that they have previously published, instead use one marker (cytb).

The authors claim a circular reasoning in dating, particularly in the Herichthys genus but also in other studies carried in the region to other groups, particularly for the case the Astyanax (Ornelas-García et al. 2008), the authors did not only used the PdM as a calibration point, thus, I encourage the authors to clearly describe the methods carried out by other authors, in order to avoid misinterpretation of the published literature.

It is not clear why the authors include the Caribbean in their discussion since the group under study is not distributed in this region, all this section is not part of the methods, and apparently the authors used previously published data to describe this section, thus it did not correspond to an original contribution.

Minor comments,
“The transmexican volcanic belt particularly is not a lava flow. It is the PdM (Hulsey et al. 2004, 2010),” : This is not the correct citation.

Perdices et al. 2002 neither Martin and Bermingam 1998, studies were not carried out on Cichlids.

Reviewer 2 ·

Basic reporting

With regards to how the information is present in the paper, there is a lot of information that sometimes is confused and probably needs a reorganization that will improve the understanding of the whole manuscript. The amount of data reported in this manuscript is very valuable but complex to organize. Just need to be rewritten in some areas. English language is very correct along the manuscript.

In my view, if I am not wrong, the paper has two main goals: (1) to understand the diversity and historical biogeography of the genus Herychthys due to its strategic geographic position relative to the Punta del Morro formation, and therefore, to testing the importance of this biogeographic event in the evolutionary history of cichlids (Huelsey et al., 2004, 2010). And (2) to evaluate the global diversification of Middle America cichlids by analysing previous on previous ddRAD or mitochondrial-dominated published data (Rican et al., 2013 and 2016). In the introduction section information is well organize (from general to particular: general context for divergence patterns and biogeography of cichlids, PdM and Herichthys as the only cichlid group north of PdM point and the group that may shed light in improve our knowledge about Middle America cichlids due to its geographic position). Nevertheless, at the end of introduction (lines 188-193), authors wrote “In this study, we thus focus on two topics; I) on species diversity analysis within Herichthys … II) on the biogeographical and temporal aspects of Herichthys diversification. For the second goal, we place these aspects of Herichthys diversification into a wider Middle America and global cichlid diversification…”. Please rewrite this last paragraph to make it more consistent with the whole introduction. However, in the Material and Methods and the Results sections information is more difficult to follow and sometimes it is not clear what data set authors refer. Results section is mainly focussed on Herichthys and PdM and no much information about global Middle America cichlids biogeography is reported. Moreover, there are some sentences in Results section that should belong to Discussion (e.g. lines 400-404; lines 417-421…). And, conversely, Discussion is also well structured and easy to follow in its present form (from particular Herichthys and PdM to global diversification of Middle America cichlids). Please, give a look at the manuscript and rewrite Results sections to organize such high and interesting amount of information. I miss also some comparison of species delimitation based on cytochrome b performed here and previous nuclear phylogenies (ddRAD; Rican et al., 2016).

Supplementary material provided is very adequate and complement results when necessary

Minor concerns:
-The title is too long, I suggest authors to shorten or rewrite it.
-Line 75: change “Tagliacolo” by “Tagliacollo”. By the way, this cite is not included in the reference section. Please include it.
-Line 99: Change “have divided” by “have been divided”
-Line 134: Change “2013,;” by “2013;”
-Line 202: “142 haplotypes”…. Line 339 “133 haplotypes”. Please review these numbers
-Lines 291-292: “The bPTP analysis was run both on the MrBayes tree and on ultrametric BEAST tree in order to compare results”. However authors do not indicate they run MrBayes in other sections of the manuscript.
-Line 386: Highlight this line as subtitle
-Line 387: Change “Fig.ure 4” by “Fig. 4”
-Lines 656 and 658: Change “t0” by “to”
-Line 659: Change “this events” by “these events”
-Figure 1. Please, include scale in the maps of Figure1.
-Figure 2. In my view, in the caption of this figure the sentence “the calibration by the Punta del Morro (PdM) is clearly an outlier (…) the PdM calibration analyses (analyses I-IV)” should be removed. I think it does not bring necessary information to this figure
-Figure 4. Letters from phylogenetic tree are not readable in printed version. For instance, in lines 451-452 authors said“… contemporaneous extinctions among the lowland genera (…) Vieja and Maskaheros (Fig. 4…), but it is very difficult to find this genera in the Figure 4.

Experimental design

This manuscript mainly constitute an original primary research that fit with the aims and scope of PeerJ. In general methodology used is adequate to solve the questions proposed by authors. Methods applied to reconstruct divergence time estimations, one of the key point of this manuscript to understand the evolutionary and biogeographic history of the Middle America cichlids, is described with sufficient detail. Information about used specimens is complete and present in Supplementary Material.

Nevertheless, focusing on analyses, I have several main concerns that should be addressed. One of these main concerns is related with phylogenetic analysis. Authors used Maximum Parsimony (MP) to infer phylogeny, however this methodology is not adequate for this purpose. MP analysis ignores branch length, associated to probability of evolutionary change and assumes that changes between nucleotide are independent (Yang, 2006), i.e. parameters of the evolutionary model cannot be incorporated in the analysis. Therefore, phylogeny may not be representing accurate relationships. I suggest authors to run Maximum Likelihood (additionally to the Bayesian Inference they carry out), in which parameter models may be incorporated, and remove Maximum Parsimony from the study.
Regarding Bayesian Inference run in BEAST, authors state “… MCMC simulations was run for 10 million generations… (Line 236)” Do authors get convergence with this number of generations? For BEAST analysis a chain of 10 millions of generations is often too short (usually enough to get convergence in MrBayes). And do authors just compare the four independent analyses they perform or do they combine them? Please clarify.
Another major concern that I have is related to molecular clock analyses in this study. I would eliminate PdM calibration in the analysis V (which is based on geological calibrations) and leave only the other two geological calibration points in order to avoid the influence of PdM calibration in the result of this analysis (line 383-385). Authors later discuss well about this circularity, and they have performed one analysis using only the PdM point as calibration (VI), which is necessary to test the biogeographic hypothesis of vicariance driven by “Punta de Morro”. Therefore, I would eliminate this analysis V. Moreover, for these BEAST analyses authors do not explain the number of generations they run for each of the four independent analyses that they later combine in LogCombiner.
An additional question that is raised regarding molecular clock analyses is the number of individuals that authors consider in these analyses. Have they included the whole data set or have they selected some individuals? This is very important to be notified because its influence in the speciation/coalescent tree prior. Authors used a “coalescent model with constant size” as prior for dating analysis (Line 246). Coalescence priors should be correctly used within a species, but authors have different species and their populations within the molecular clock analysis. Therefore, coalescent tree prior may bias results, as well as using a speciation prior, which could be overestimating speciation events. The best way for integrating both scales (species and populations) in BEAST software is to use the option STARBEAST (Heled & Drummond, 2010) that takes into account coalescence within the speciation (Yule) process and that render both species-tree and gene trees. Nevertheless, although possible, this approach is more adequate for multilocus dataset than for only one gene (cytb) due to error in date estimations may be higher in one-locus analysis. STARBEAST also tend to provide more recent divergence times than a “standard beast” when estimates the species-tree. For this reason I recommend authors to run STARBEAST for the six calibration analyses using the partition by codon cyt b and discuss results of the cytb b gene tree obtained from these analyses, which would have incorporated coalescence within the speciation process.

Validity of the findings

The paper of Perez-Miranda et al. is a great contribution to the knowledge of the evolutionary patterns and biogeography of freshwater fish of Middle America and Caribbean Region. The debate about the colonization of Middle America - Caribbean region and the relationships of their biota is still open, especially focus on when started the presence of permanent emerged lands in this region (e.g., among others, GAARlandia hypothesis; Iturralde-Vinent and MacPhee 1999; Iturralde-Vinent, 2006…) and authors provide here, with this study about the cichlid genus Herichthys, a new piece of this interesting puzzle. This paper also highlights the importance of having several accurate calibration points to obtain a robust historical biogeography of any particular taxonomic group. This paper deserves to be published in PeerJ after authors deal with major concerns I have and that I have previously reported. Therefore, I encourage authors to deal with these major concerns in order to make suitable this manuscript for publication.

---

## Round 0.2 · Minor Revisions

Dear Authors,

It took a bit longer over the holiday period, but the review of your revised MS are in. I have received a recommendation of minor revision, and I agree with this assessment. In my opinion the MS is almost ready to be accepted.

I agree with the reviewer in focusing on STARBEAST analyses which permit to mix intraspecific and interspecific variation within a phylogenetic inference framework without, in principle, inflating the ages of recent speciation events or MRCA estimates within species.
Second, I would appreciate if you justified the use of the strict molecular clock prior in you STARBEAST analyses. I do not recall that you tested for clock-like behavior of your data, and permitting non-clock like rate of molecular evolution is not computationally costly, so I am not sure why you chose the strict molecular clock prior.
Finally, there are still typos throughout the MS, something that a spell checker should easily detect, so please try to pick up these small problems before resubmitting the final version.
Job well done, and I look forward to receiving your revision shortly.

Sincerely,

Tomas Hrbek

Reviewer 2 ·

Basic reporting

This is the second time that I have reviewed this manuscript and I have to say it has been improved a lot by authors. They have solved my main concerns about dating analyses and calibration points, as they now have analyzed "Punta de Morro" calibration alone in order to compare with the remaining calibrations. The inclusion of PdM along with other calibration points within the same analysis in the previous version of the manuscript was biasing results and made difficult to extract accurate conclusions from them. In fact, now "Methods" section is clearer and well-organized than before. At the same time, now it is also clear to me how authors incorporate the Rad-seq tree topology of Oldrich et al 2016 in the biogeographic analysis, which was not easy to follow in the previous version of the manuscript. For this reason, I consider the manuscript is now prepared to be accepted in PeerJ. Although I still have some minor concerns regarding analyses that I explain in the "Experimental design" section and that should be revised. Authors also should do a proofreading of the manuscript as I have found some typing mistakes.

Experimental design

This manuscript constitutes a primary research that fit in the Aims and Scope of the journal and provide a robust background about the diversification of cichlids and particularly for the genus Herichthys. In general methodology used is correct and adequate to solve the proposed evolutionary questions. I just still have a minor concern about the selection of the evolutionary model used for phylogenetic performance. Authors say that they used the GTR+G+I model, but they do not specify any software to do that, such as jModeltest of PartitionFinder. Why did authors use this model? Is it because it is the most complex model? There is a debate about the use of the most complex model and the overestimation of parameters if this model is not the adequate for the data or the non influence of the complexity of the model according to different authors. In my experience and knowing this fish group the selection of the model is not going to affect strongly to results, it would be more elegant that authors estimate themselves the best-fit model for these data. The same is about the use of Beast, a coalescent model is not adequate when different species are included in the analyses, and would be more correct to used an speciation prior (Yule or Birth date), taking into account that in that case just one sequence per species should be keeping in the analyses. However, using StarBeast is correct when the data are constituted by species and populations (or several individuals per species), as is the case of this study, due to this approach is able to integrate speciation and coalescent processes in the same analyses. I do not understand either why authors used a relaxed molecular clock prior in Beast and a strict clock in StarBeast, have the tested this statement?. Please, clarify. My recommendation is that authors used only StarBeast analyses (even if they only have the cytochrome b marker), in this case they can estimate the species tree and the gene tree at the same time. These are the minor concerns that I have for this manuscript and those that should be taking into account before the acceptance of the manuscript.

Validity of the findings

This manuscript is of interest to the scientific community due to the importance of cichlid fish as model for evolutionary and biogeographic studies. The manuscript is also testing the influence of a biogeographical barrier that have been consider as promotor of vicariant patterns in different organisms. The finding of this study suggest an older diversification for cichlids in comparison to other studies and authors provide robust data for support this conclusion, which is also well developed in the Discussion section. They also provide a more complete phylogeny for the genus Herichthys, showing the need of a systematic review of the group.

---

## Round 0.3 · accepted · Accept

Dear authors,

Thank you for submitting your revision. I am happy with your revision, and so I am recommending to accept your MS in its current form.
Congratulations on a very nice MS, and a job well done.

Sincerely,
Tomas Hrbek